# Prediction of suicidal ideation risk in a prospective cohort study of medical interns

Tyler L. Malone[1,2¤a], Zhou Zhao[3], Tzu-Ying Liu[1¤b], Peter X. K. Song[1], Srijan Sen[3‡*], Laura J. Scott[1,2‡*]

**1** Department of Biostatistics School of Public Health, University of Michigan, Ann Arbor, Michigan, United States of America, **2** Center for Statistical Genetics, University of Michigan, Ann Arbor, Michigan, United States of America, **3** Michigan Neuroscience Institute, University of Michigan, Ann Arbor, Michigan, United States of America

¤a Current address: Department of Health Policy and Management, University of North Carolina, Chapel Hill, North Carolina, United States of America
¤b Current address: Biogen Inc., Cambridge, Massachusetts, United States of America
‡ These authors are joint senior authors on this work
* srijan@med.umich.edu (SS); ljst@umich.edu (LJS)

**Data Availability Statement:** The minimal data set underlying the results described in our paper is available through the Inter-university Consortium for Political and Social Research (ICPSR) at the

## Abstract

The purpose of this study was to identify individual and residency program factors associated with increased suicide risk, as measured by suicidal ideation. We utilized a prospective, longitudinal cohort study design to assess the prevalence and predictors of suicidal ideation in 6,691 (2012–2014 cohorts, training data set) and 4,904 (2015 cohort, test data set) first-year training physicians (interns) at hospital systems across the United States. We assessed suicidal ideation two months before internship and then quarterly through intern year. The prevalence of reported suicidal ideation in the study population increased from 3.0% at baseline to a mean of 6.9% during internship. 16.4% of interns reported suicidal ideation at least once during their internship. In the training dataset, a series of baseline demographic (male gender) and psychological factors (high neuroticism, depressive symptoms and suicidal ideation) were associated with increased risk of suicidal ideation during internship. Further, prior quarter psychiatric symptoms (depressive symptoms and suicidal ideation) and concurrent work-related factors (increase in self-reported work hours and medical errors) were associated with increased risk of suicidal ideation. A model derived from the training dataset had a predicted area under the Receiver Operating Characteristic curve (AUC) of 0.83 in the test dataset. The suicidal ideation risk predictors analyzed in this study can help programs and interns identify those at risk for suicidal ideation before the onset of training. Further, increases in self-reported work hours and environments associated with increased medical errors are potentially modifiable factors for residency programs to target to reduce suicide risk.

## Introduction

Physicians may be at elevated risk for suicide compared to the general population [1,2]. Recent suicides have raised concerns that training physicians may be at particularly high risk [3,4].

University of Michigan (https://www.openicpsr.org/openicpsr/project/129225/version/V1/view).

**Funding:** SS reports having received research funding from the NIMH (R01 MH101459; website: https://www.nimh.nih.gov/index.shtml) and an American Foundation for Suicide Prevention Standard Research Grant (website: https://afsp.org/). The funders had no role in study design, data collection and analysis, decision to publish, or preparation of the manuscript.

**Competing interests:** I have read the journal's policy and the authors of this manuscript have the following competing interests: SS reports having received research funding from the NIMH (R01 MH101459; website: https://www.nimh.nih.gov/index.shtml) and an American Foundation for Suicide Prevention Standard Research Grant (website: https://afsp.org/). The funders had no role in study design, data collection and analysis, decision to publish, or preparation of the manuscript. The above competing interests do not alter our adherence to PLOS ONE policies on sharing data and materials.

Indeed, levels of suicidal ideation are elevated in residents and appear to increase dramatically with the onset of training [5]. Growing recognition of resident suicide and poor mental health have led educational leaders, residents and medical organizations to call for interventions and changes in the training system [3,6–12]. With this goal in mind, identification of at-risk individuals can help with the development of effective interventions and structural changes to reduce in-training physician suicide.

A theoretical understanding of suicide risk is important to effectively identify at-risk physicians. Joiner's Interpersonal Theory of Suicide [13] summarizes suicide risk as a function of an individual's desire and capability for suicide [14,15]. Specifically, the Interpersonal Theory states that suicide desire is related to thwarted belongingness (i.e., the need to belong) and perceived burdensomeness (i.e., perceptions of personal incompetence or liability). Experienced individually, thwarted belongingness and perceived burdensomeness are proposed as proximal and sufficient causes of passive suicidal ideation (e.g., "I wish I was dead" or "I would be better off dead"). When experienced together, thwarted belongingness and perceived burdensomeness can lead to active suicidal ideation, or an active desire for suicide (e.g., "I want to kill myself"). Suicide desire, in turn, can lead to suicide attempts, depending on the degree of suicide capability [14,15].

Cornette et al. [16] concluded that the Interpersonal Theory was consistent with existing evidence on physician suicide. In particular, the authors highlighted several factors that could predispose training physicians to experience increased risk of thwarted belongingness or perceived burdensomeness, including academic burnout, financial debt, emotional distress, social isolation, and an excessive sense of responsibility for patients' health outcomes. In addition, the authors posited that medical training, and the accompanying exposure to patients, acclimates students to pain and injury. Combined with the additional knowledge of lethal medication dosing, physicians are possibly more likely to acquire suicide capability. In addition to work by Cornette et al. [16], follow-up studies by Fink-Miller [17] and Loas et al. [18] (among others [19,20]), have also supported the applicability of the Interpersonal Theory to physicians. Despite this previous research, consensus empirical evidence of risk factors among physicians, particularly physicians in training, is lacking. A recent systematic review of medical student suicide rates found a low number of available studies and insufficient data to complete a formal meta-analysis [21]. In particular, the authors noted the critical need for additional empirical research on suicide risk factors among medical interns.

Given the need for additional empirical evidence on intern suicide, the objectives of this study were to (1) estimate suicide risk among training physicians, and (2) using insight from the Interpersonal Theory of Suicide, empirically assess individual and residency program factors proposed to drive the development of suicide risk, as measured by suicidal ideation.

## Methods

### Study design, setting, and participants

The Intern Health Study is a multi-institutional prospective, longitudinal cohort study that annually assesses training physicians as they transition into residency training [5,22,23]. Individuals are sent an e-mail invitation to participate in the Intern Health Study approximately two to three months prior to commencing internship. Potential participants are informed that the Intern Health Study analyzes biological and program factors involved in the development of depression under stress, and that the results from the study will be used to improve the residency experience and provide important information about physician health. Upon agreeing to join the study (through the provision of electronic consent), participants complete an online baseline survey approximately one to two months prior to commencing internship.

Participants then complete additional online follow-up surveys during months 3, 6, 9, and 12 of their internship year (designated as 1st, 2nd, 3rd, and 4th quarter, respectively; participants have approximately one month to complete each of the quarterly surveys). Our research focused on Intern Health Study participants entering residency programs across specialties in the 2012–2013 (218 hospital systems), 2013–2014 (243 hospital systems), 2014–2015 (113 hospital systems), and 2015–2016 (366 hospital systems) academic years (designated as the 2012 cohort, 2013 cohort, 2014 cohort, and 2015 cohort, respectively). Individuals were given $50 in gift certificates to participate in the study. The study design was approved by the Institutional Review Board at the University of Michigan and the participating hospitals in the Intern Health Study (IRB Number: HUM00033029; First Approved: 07/2009).

## Survey data

All survey data on outcomes and predictors of interest were collected through a secure online website designed to maintain confidentiality, with subjects identified only by numeric IDs. No links between the identification number and the subjects' identities were maintained.

Given the importance of suicidal ideation in the Interpersonal Theory of Suicide and the challenges in collecting data on the rarer outcomes of suicide attempts and fatalities [15], the outcome variable for our research was the presence of suicidal ideation during internship. In addition, predictor variables of interest included self-report measures with hypothesized or observed effects on an individual's sense of belonging (e.g., marital status, number of children, neuroticism) [15,16], perceived burdensomeness (e.g., medical errors, work hours) [16,24,25], and/or acquired suicide capability (e.g., previous suicidality, early family environment, stressful life events, medical specialty) [14–16,24]. Other predictors such as depressive symptoms, anxiety symptoms, sex, age, race and ethnicity, and sleep hours have demonstrated empirical associations with suicidal behavior and/or are commonly included as covariates in models of suicidal behavior [15,26–28]. Thus, we included these predictors in our study as well.

As mentioned above, the Interpersonal Theory states that suicidal ideation is a function of thwarted belongingness and perceived burdensomeness. In contrast, acquired suicide capability is not proposed to directly affect suicidal ideation, but instead affects the development of suicidal intent and the likelihood of suicide attempts and fatality. However, we decided to include predictors with an observed or hypothesized effect on acquired suicide capability for two reasons. First, if increased suicide capability leads to suicide attempts, then the trauma of a suicide attempt could also plausibly lead to increased suicidal ideation [15]. Second, the Interpersonal Theory states that individuals with suicidal ideation *and* suicide capability are at higher risk for a suicide fatality [15]. Thus, if factors that are predictive of acquired suicide capability are also predictive of suicidal ideation, then this observation would be clinically relevant. Given this rationale, we decided to include survey data on the aforementioned predictors of acquired suicide capability.

The baseline survey assessed suicidal ideation and depressive symptoms over the past two weeks through the Patient Health Questionnaire-9 (PHQ-9) [29]. For each item on the PHQ-9, interns indicated whether, during the previous two weeks, the listed symptom had bothered them "not at all," "several days," "more than half the days," or "nearly every day," with the responses scored as 0, 1, 2, or 3, respectively. We measured suicidal ideation through a positive response to the ninth item of the PHQ-9, "Thoughts that you would be better off dead or hurting yourself in some way" during the previous two weeks (i.e., we dichotomized the ninth item such that a score of 0 indicated no suicidal ideation and a score of 1, 2, or 3 indicated suicidal ideation). A positive response to this item increases the cumulative risk for a suicide attempt or fatality over the next year by 10- and 100-fold, respectively [30]. Depressive symptoms were measured using the sum of the first eight items (PHQ-8) of the Patient Health Questionnaire

[29]. The sum of the PHQ-8 responses, when dichotomized as a score less than 10 or greater than or equal to 10, has high sensitivity and specificity for the diagnosis of major depressive disorder (MDD), [31,32] with a diagnostic validity comparable to clinician-administered assessments [31].

In addition, the baseline survey assessed anxiety symptoms over the past 2 weeks through the General Anxiety Disorder-7 (GAD-7), a reliable and valid measure of anxiety in psychiatric [33] and general population samples [34]. The personality trait of neuroticism was assessed at baseline through the NEO-Five Factor Inventory (NEO-FFI), [35] and early family environment stress was assessed through the Risky Families Questionnaire [36]. The baseline survey also collected data on personal history of depression, exposure to recent stressful life events, and intern demographics.

The quarterly follow-up surveys assessed interns again for their self-report in the past two-week experience of suicidal ideation, PHQ-8 depressive symptoms, GAD-7 anxiety symptoms, and stressful life events, as well as work hours and average sleep hours in the past week, and medical errors in the last three months.

## Statistical methods

To identify predictors of suicidal ideation during internship, we first split our data into two groups, interns from the 2012–2014 cohorts and interns from the 2015 cohort. We used data from the 2012–2014 cohorts as a "training" dataset to fit a logistic mixed effects model with random intercepts [37,38]. Random intercepts were specified to account for repeated measurements of interns over the course of internship. Our model used variables from baseline and follow-up to estimate an intern's risk of suicidal ideation during a particular quarter of internship (quarters 2, 3 and 4). We selected predictors (or fixed effects) using backward elimination with an α-to-remove value of 0.10 (Wald-type test [39]). We chose to use backward elimination to balance model interpretability (parsimony) with the predictive ability of our model.

An intern could contribute up to three quarters of outcome data (quarters 2, 3 and 4). For each outcome quarter, we included the set of interns that had complete phenotype data for all variables of interest at (1) baseline and (2) a consecutive set of prior and current quarters (i.e., 1st and 2nd, 2nd and 3rd, and/or 3rd and 4th quarters). Thus, interns without complete baseline data and complete data for at least one consecutive set of two quarters were excluded from further analysis. Our mixed effects model will provide valid estimation and inference when missing data are missing at random (MAR). Before beginning analysis, we assessed if our complete-case data met the MAR assumption using longitudinal plots stratified by missing patterns and logistic regression models of missing indicators for covariates [40]. We found no evidence that the complete case data violated the MAR assumption [41].

After fitting our model with the training dataset, we used the logistic regression model with fixed effects to predict suicidal ideation among interns in the 2015 cohort (i.e., the "test" dataset). In comparison to an internal cross-validation approach, our use of training and test samples from different cohort years allowed us to more rigorously evaluate the external validity of the prediction model [42,43]. We assessed the predictive ability of our model using a Receiver Operating Characteristic (ROC) curve and estimation of area under the curve (AUC) [44]. An AUC value of 0.5 is the expected discriminatory ability of a model that discriminates subjects randomly, values of 0.7 to 0.8 are generally considered acceptable, and values above 0.8 are generally considered good [44].

We conducted analyses using SAS software version 9.4 (SAS Institute Inc., Cary, North Carolina, United States of America). R version 3.3.2 was used to create additional figures (R Foundation for Statistical Computing, Vienna, Austria).

## Results

We sent study invitations via e-mail to 6,691 interns from the 2012–2014 cohorts (323 hospital systems) and 4,904 interns from the 2015 cohort (366 hospital systems). For 117 interns, our e-mail invitations were returned as undeliverable and we were unable to obtain a valid e-mail address. Of the remaining invited interns, 59.4% agreed to participate in the study (3,896 interns from 2012–2014 training set cohorts and 2,920 interns from the 2015 test set cohort). Among the training set, 2,293 interns had complete information at baseline and for one ($n = 480$), two ($n = 347$) or three ($n = 1,466$) sets of consecutive quarters (i.e., 5,572 complete consecutive quarter observations). Among the test set, 2,043 interns had complete information at baseline and for one ($n = 398$), two ($n = 254$), or three ($n = 1,391$) sets of consecutive quarters (i.e., 5,079 complete consecutive quarter observations).

Table 1 provides baseline characteristics of study participants. The mean age of analyzed interns was 27.4 years (standard deviation = 2.7 years), 50.7% were female, 65.2% were white, 19.6% Asian, 2.8% Latino, and 3.2% African American. The most common specialties were internal medicine (28.4%), pediatrics (12.4%), and surgery (9.3%). Of interns, 60.2% were single, 39.0% were engaged or married, and 7.6% had children. Interns had an average baseline depressive symptoms score of 2.5 (out of 27), anxiety symptoms score of 2.8 (out of 21), neuroticism score of 21.0 (out of 56) and early family environment score of 12.4 (out of 65). Slightly less than half, 45.1% of interns indicated a personal history of depression, 27.7% experienced one or more self-reported stressful life events at baseline, and 3.0% had suicidal ideation.

Fig 1 shows changes in the prevalence of reported suicidal ideation during internship. At the 1st quarter of internship, 6.1% of interns reported suicidal ideation (up from 3.0% at baseline). The prevalence at the 2nd, 3rd, and 4th quarters was 7.8%, 6.9%, and 6.6%, respectively. 16.4% of interns reported suicidal ideation at least once during internship. Interns with baseline suicidal ideation had much higher prevalence of reported suicidal ideation throughout the internship (average 44.7% suicidal ideation) than interns without baseline suicidal ideation (average 5.7% suicidal ideation).

Table 2 shows results from the multiple regression of suicidal ideation for the 2012–2014 cohorts training set intern characteristics, estimated using a logistic mixed effects model of baseline, prior and current quarter data (in addition, see S1 Table in the online supplement, which provides a complementary univariable analysis of baseline intern characteristics and suicidal ideation). We found that prior quarter suicidal ideation (Odds Ratio (OR) = 7.84, $p = 4.4 \times 10^{-32}$, baseline suicidal ideation (OR = 5.41, $p = 2.5 \times 10^{-11}$), increase in self-reported work hours from the previous quarter (OR = 1.34, $p = 4.0 \times 10^{-6}$), current quarter self-reported medical errors (OR = 1.80, $p = 1.1 \times 10^{-5}$), prior quarter depressive symptoms score (OR = 1.36, $p = 1.8 \times 10^{-4}$), baseline neuroticism score (OR = 1.33, $p = 3.4 \times 10^{-4}$), and baseline personal history of depression (OR = 1.45, $p = 4.7 \times 10^{-3}$) were significant predictors of current quarter suicidal ideation under a threshold of $p < .01$, holding all other model covariates constant. Table 2 shows additional predictors that were significant under less strict thresholds of $p < .05$ and $p < .10$, including baseline anxiety score. Notably, a higher baseline anxiety score was associated with a lower odds ratio of suicidal ideation (OR = 0.86, $p = 0.03$), holding all other model covariates constant. However, when analyzed on its own, baseline anxiety score was significantly and positively associated with suicidal ideation (see S1 Table in the online supplement).

To assess the predictive ability of our full model in a separate set of interns, we used the fixed effects model estimates to predict suicidal ideation among interns in the 2015 cohort test set. The AUC for the full model was 0.83, indicating that, based on the model with baseline,

**Table 1. Baseline characteristics of Intern Health Study participants entering residency programs across specialties in the 2012–2014 (*n* = 2,293) or 2015 (*n* = 2,043) academic years.**

| | All Interns | Training Set | Test Set |
|---|---|---|---|
| **Number of Interns** | 4,336 | 2,293 | 2,043 |
| **Number of Observations** | 10,651 | 5,572 | 5,079 |
| *Mean (Standard Deviation)* | | | |
| **Age, Years** | 27.4 (2.7) | 27.5 (2.6) | 27.4 (2.7) |
| **Depressive Symptoms Score**[a] | 2.5 (2.9) | 2.5 (2.8) | 2.5 (2.9) |
| **Anxiety Symptoms Score**[b] | 2.8 (3.3) | 2.7 (3.1) | 2.9 (3.4) |
| **Neuroticism Score**[c] | 21.0 (8.7) | 20.9 (8.5) | 21.1 (8.8) |
| **Early Family Environment Score**[d] | 12.4 (9.0) | 12.2 (9.0) | 12.5 (9.0) |
| *No. (Percent of Sample)* | | | |
| **Sex** | | | |
| Male | 2,138 (49.3%) | 1,142 (49.8%) | 996 (48.8%) |
| Female | 2,198 (50.7%) | 1,151 (50.2%) | 1,047 (51.3%) |
| **Race/Ethnicity** | | | |
| White | 2,827 (65.2%) | 1,460 (63.7%) | 1,367 (66.9%) |
| African American | 137 (3.2%) | 63 (2.8%) | 74 (3.6%) |
| Latino | 123 (2.8%) | 74 (3.2%) | 49 (2.4%) |
| Asian | 851(19.6%) | 492 (21.5%) | 359 (17.6%) |
| Other | 398 (9.2%) | 204 (8.9%) | 194 (9.5%) |
| **Specialty** | | | |
| Internal Medicine | 1,231 (28.4%) | 729 (31.8%) | 502 (24.6%) |
| Surgery | 402 (9.3%) | 246 (10.7%) | 156 (7.6%) |
| OB/GYN | 236 (5.4%) | 104 (4.5%) | 132 (6.5%) |
| Pediatrics | 538 (12.4%) | 281 (12.3%) | 257 (12.6%) |
| Psychiatry | 247 (5.7%) | 142 (6.2%) | 105 (5.1%) |
| Emergency Medicine | 343 (7.9%) | 180 (7.9%) | 163 (8.0%) |
| Family Practice | 267 (6.2%) | 112 (4.9%) | 155 (7.6%) |
| Other | 1,072 (24.7%) | 499 (21.8%) | 573 (28.1%) |
| **Marital Status** | | | |
| Single | 2,609 (60.2%) | 1,401 (61.1%) | 1,208 (59.1%) |
| Engaged/Married | 1,692 (39.0%) | 873 (38.1%) | 819 (40.1%) |
| Separated/Divorced | 35 (0.8%) | 19 (0.8%) | 16 (0.8%) |
| **Has Children** | 328 (7.6%) | 158 (6.9%) | 170 (8.3%) |
| **Suicidal Ideation** | 131 (3.0%) | 72 (3.1%) | 59 (2.9%) |
| **Personal History of Depression** | 1,954 (45.1%) | 1,027 (44.8%) | 927 (45.4%) |
| **One or More Stressful Life Events** | 1,200 (27.7%) | 673 (29.4%) | 527 (25.8%) |

*Notes*: Training set is comprised of interns from the 2012–2014 cohorts. Test set is comprised of interns from the 2015 cohort. Intern characteristics were self-reported. *Abbreviations*: OB/GYN = obstetrics and gynecology.

[a]Assessed via the Patient Health Questionnaire-8.

[b]Assessed via the 7-item General Anxiety Disorder-7.

[c]Assessed via the NEO-Five Factor Inventory.

[d]Assessed through the Risky Families Questionnaire.

prior, and current predictors (full model), a randomly chosen intern with suicidal ideation has an 83% probability of having a higher predicted risk of suicidal ideation than a randomly chosen intern without suicidal ideation (Fig 2).

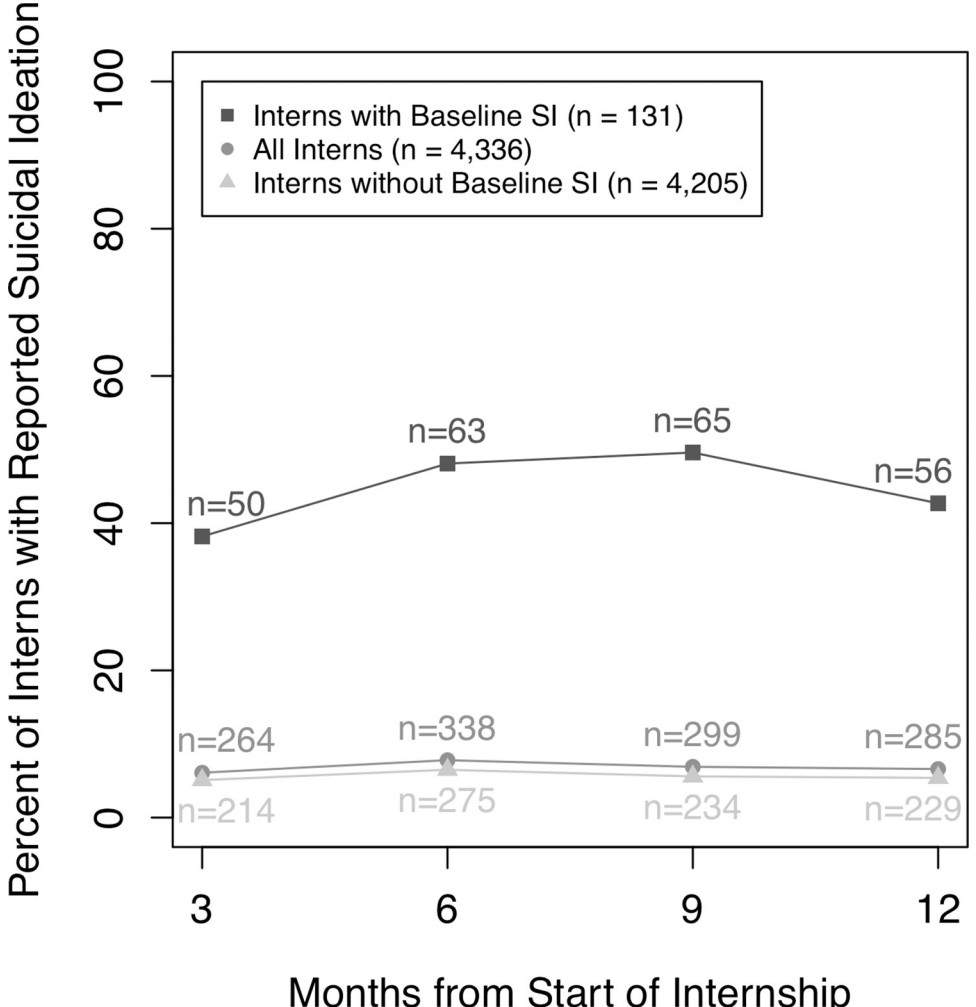

**Fig 1. Prevalence of reported suicidal ideation by quarter of internship and baseline intern suicidal ideation status.** Prevalence rates calculated among interns in the 2012–2015 cohorts (*n* = 4,336). SI = suicidal ideation.

To assess the predictive value of models without prior and/or current quarters data, we fit models restricted to (1) the baseline predictors or (2) the baseline and the prior quarter (i.e., the quarter before measurement of suicidal ideation) predictors. The AUC for the baseline and prior quarter predictors model was 0.82 (Fig 2), and the ORs were similar to the full model (Fig 3). In contrast, the AUC for the baseline predictors model (no prior quarter data) was lower than the full model, 0.75 (Fig 2), and the ORs for suicidal ideation were higher than the full model for many variables, including baseline suicidal ideation, baseline personal history of depression, baseline depressive symptom score, and baseline neuroticism (Fig 3). We further assessed the predictive value of a full model that does not include baseline or prior quarter suicidal ideation as predictor variables. We found an AUC of 0.79 (Fig 2) and, compared to the full model, the ORs for suicidal ideation where higher for many variables, including baseline personal history of depression, baseline depressive symptoms score, baseline neuroticism score, prior quarter depressive symptoms score, and current quarter self-reported medical errors (Fig 3).

To illustrate the distribution of predicted suicidal ideation risk in 2015 cohort individuals with suicidal ideation, we plotted the distribution of predicted suicidal ideation risk for each

**Table 2. Logistic mixed effects multiple regression analysis predicting current quarter suicidal ideation from intern mental health, demographics, and internship characteristics.**

| | OR[a] | 95% CI | p |
|---|---|---|---|
| *Baseline Characteristics* | | | |
| **Suicidal Ideation** | 5.41 | (3.30–8.88) | $2.5 \times 10^{-11}$ |
| **Neuroticism Score**[b] | 1.33 | (1.14–1.55) | $3.4 \times 10^{-4}$ |
| **Personal History of Depression** | 1.45 | (1.12–1.87) | $4.7 \times 10^{-3}$ |
| **Male Sex** | 1.39 | (1.09–1.78) | 0.01 |
| **Depressive Symptoms Score**[c] | 1.15 | (1.01–1.30) | 0.03 |
| **Anxiety Score**[d] | 0.86 | (0.74–0.99) | 0.03 |
| *Prior Quarter Characteristics* | | | |
| **Suicidal Ideation** | 7.84 | (5.59–11.00) | $4.4 \times 10^{-32}$ |
| **Depressive Symptoms Score**[c] | 1.36 | (1.16–1.59) | $1.8 \times 10^{-4}$ |
| **Anxiety Score**[d] | 1.17 | (1.01–1.36) | 0.04 |
| *Current Quarter Characteristics* | | | |
| **Increase in Work Hours from Prior Quarter** | 1.34 | (1.19–1.52) | $4.0 \times 10^{-6}$ |
| **One or More Medical Errors**[e] | 1.80 | (1.38–2.33) | $1.2 \times 10^{-5}$ |
| **One or More Stressful Life Events** | 1.25 | (0.96–1.62) | 0.1 |
| **Average Sleep Hours** | 0.88 | (0.78–1.00) | 0.04 |
| **Months since Start of Internship** | 0.95 | (0.91–1.00) | 0.05 |

*Notes*: Intern variables were self-reported. Baseline variables were known at the beginning of internship, prior quarter variables describe characteristics three months prior to the time of outcome, and current quarter variables describe intern characteristics at the time of outcome. Model variables were selected through backward elimination using an α-to-remove value of 0.1. *Abbreviations*: OR = odds ratio; CI = confidence interval.

[a]For continuous variables other than "Months since Start of Internship", the odds ratio represents the change in the odds of suicidal ideation associated with a one standard deviation increase from the mean of the independent variable.

[b]Assessed via the NEO-Five Factor Inventory.

[c]Assessed via the Patient Health Questionnaire-8.

[d]Assessed via the General Anxiety Disorder-7.

intern observation (see S1 Fig in the online supplement). Although, as expected, a large proportion of individuals with current suicidal ideation had relatively high predictive risk scores, some individuals with current suicidal ideation had very low predicted risk. To see if the predictive ability of the full model differed in individuals without a baseline report of suicidal ideation, we excluded interns with baseline suicidal ideation from the 2012–2014 cohorts and refit the model; the AUC was 0.80. This model generally had similar magnitudes of effect for the included predictor variables compared to the effect sizes for the full 2012–2014 cohort training model.

## Discussion

The objectives of our research were to (1) estimate suicide risk among training physicians, and (2) empirically assess individual and residency program factors proposed to drive the development of intern suicide risk, as detailed by the Interpersonal Theory of Suicide [13]. To this end, our multi-site longitudinal cohort study identified a substantial increase in suicidal ideation as soon as the internship started, with 16.4% of training physicians reporting suicidal ideation over the course of the year. In addition, we identified a set of individual factors present before internship that predicted future suicidal ideation with fair to good accuracy based on

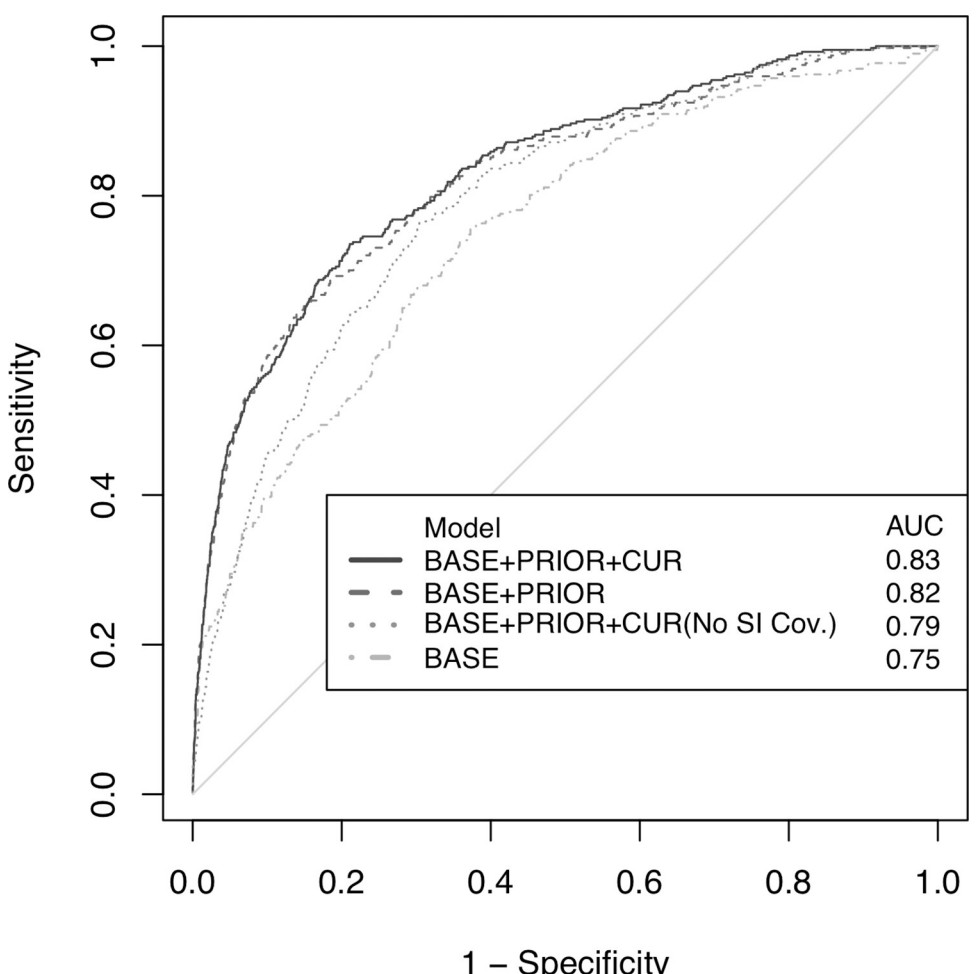

**Fig 2. Receiver Operating Characteristic curves for prediction models of suicidal ideation during internship.**
*Notes*: Receiver Operating Characteristic curves were calculated for the 2015 cohort test set ($n$ = 2,043) by applying prediction models constructed from the 2012–2014 cohorts training set ($n$ = 2,293). The grey reference diagonal line represents the area under the curve value (AUC) of 0.50 (the expected discriminatory ability of a model that discriminates subjects randomly); SI = suicidal ideation; BASE = Model includes baseline predictors of suicidal ideation; BASE+PRIOR = Model includes base + prior quarter predictors of suicidal ideation; BASE+PRIOR +CUR = Model includes base+prior+current quarter predictors of suicidal ideation; No SI Cov. = Model includes base +prior+current quarter predictors of suicidal ideation except for baseline and prior quarter suicidal ideation.

AUC. We also identified program level factors present during training that were associated with increased prediction accuracy. We discuss the key results below, including their relevance within the context of Joiner's Interpersonal Theory of Suicide.

The two-fold increase in suicidal ideation during internship was a key finding of our research. As detailed by Cornette et al. [16], there are multiple factors that could predispose training physicians to experience increased risk of suicidal ideation, including academic burn-out, financial debt, emotional distress, social isolation, and an excessive sense of responsibility for patients' health outcomes. The increase in suicidal ideation among training physicians is particularly concerning given that physicians' exposure to patients and knowledge of lethal medication dosing could also indicate increased suicide capability [16,19]. As described by the Interpersonal Theory of Suicide, combined suicidal ideation and capability greatly increases the risk of suicide attempts and fatality [15]. Thus, the prevalence of suicidal ideation within

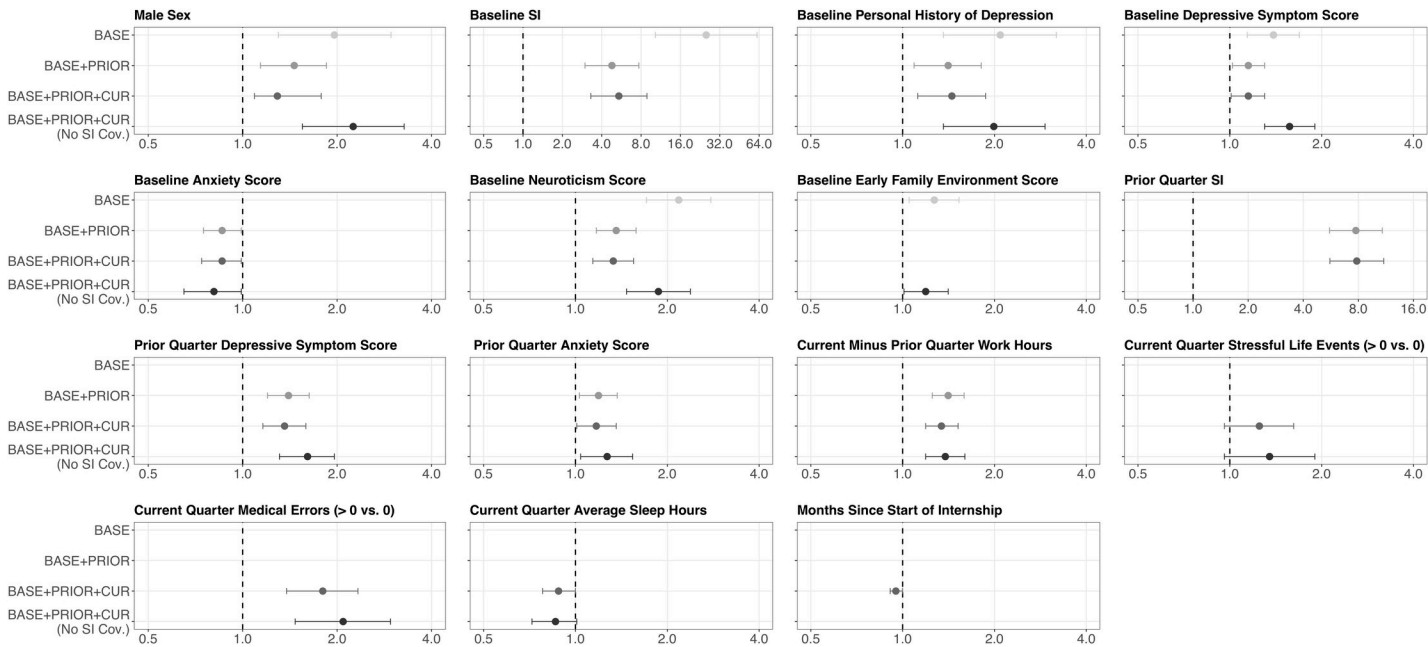

**Fig 3. Logistic mixed effects multiple regression models predicting suicidal ideation during internship.** *Notes*: Models include observations from the 2012–2014 cohorts (*n* = 2,293). Intern variables were self-reported. BASE = Model includes baseline predictors of suicidal ideation; BASE+PRIOR = Model includes base + prior quarter predictors of suicidal ideation; BASE+PRIOR+CUR = Model includes base+prior+current quarter predictors of suicidal ideation; No SI Cov. = Model includes base+prior+current quarter predictors of suicidal ideation except for baseline and prior quarter suicidal ideation; SI = suicidal ideation. For continuous variables other than "Months since Start of Internship," the odds ratio represents the change in the odds of suicidal ideation associated with a one standard deviation increase from the mean of the independent variable. Depressive symptom score was assessed via the Patient Health Questionnaire-8. Anxiety symptom score was assessed via the General Anxiety Disorder-7. Neuroticism score was assessed via the NEO-Five Factor Inventory. Early Family Environment score was assessed via the Risky Families Questionnaire.

our study sample underscores the magnitude of poor mental health and suicide risk among training physicians and the need for systemic reform that creates a healthier work environment. Connecting training physicians experiencing suicidal ideation to appropriate clinical services is an important next step and shows promise in reducing suicide prevalence [7,8,15,23,45,46].

Another key finding of our research demonstrated that previous suicidal ideation was a strong risk factor for current suicidal ideation. First, this result suggests that the factors that predispose to suicidal ideation before internship continue to predispose to suicidal ideation during internship. Second, this result is consistent with mechanisms proposed by the Interpersonal Model, which suggests that previous suicidality is predictive of current suicidality [15]. Despite the predisposing effect of past suicide ideation on current suicide ideation, when baseline and prior suicidal ideation were removed from the full model, the model retained good predictive ability (i.e., the AUC decreased only slightly, from 0.83 to 0.79). Furthermore, factors that were significant in the full model had stronger effect sizes once baseline and prior suicidal ideation were removed, suggesting that the remaining factors were capturing additional information on the underlying risk of suicidal ideation.

Self-reported medical errors were also predictive of suicidal ideation. Previous research has observed associations between medical errors and feelings of shame and guilt among physicians [47,48]. Thus, medical errors could plausibly be related to suicidal ideation through increased perceptions of burdensomeness [15]. More generally, the findings on medical errors highlight the possible effects of past trauma on current suicidal ideation. Previous literature has consistently described links between previous traumatic experiences and current suicidal

behavior among medical residents [49,50]. Going forward, our finding suggests that structural changes to decrease medical errors, such as increased supervision and standardized checklists, may improve both patient safety and physician safety through the reduction of "second victim syndrome" [51]. In addition, teaching physicians how to effectively cope with medical errors could be helpful in reducing the downstream effects of errors on suicidal ideation [9]. Another possibility is that suicidal ideation and medical errors are caused by a third confounding factor such as physician burnout or poor emotion regulation. In the case of physician burnout, the use of Schwartz rounds [52] or other approaches to reduce burnout could be effective options to reduce both suicidal ideation and medical errors. Recent research suggests that individuals with previous traumatic experiences, including adverse childhood experiences, could be particularly vulnerable to burnout [53] and thus more likely to benefit from interventions.

In addition to burnout, differences between interns in emotion regulation and stress could explain several other results from the current research. For example, changes in self-reported work hours and decreased sleep hours were both associated with increased risk of suicidal ideation. Increased psychological distress due to increased work hours [54,55] or reduced sleep hours [56] plausibly leads to negative affect and increased burdensomeness or thwarted belongingness [15,16]. Neuroticism, another statistically significant predictor of suicidal ideation in our analysis, can also lead to negative affect and increased feelings of hopelessness [57]. According to the Interpersonal Theory of Suicide, hopelessness is particularly relevant as a determinant of active suicidal ideation [15]. Moreover, depression has been consistently linked to suicidal ideation, both in the current study and previous research [15]. Prior analyses suggest that individuals predisposed to depression are more likely to experience negative affect [58], and that depression increases the desire for suicide [15]. Importantly, the current study demonstrates that simple screening tools, such as the PHQ-9 [29] and NEO-FFI [35], can be used to assess key components of intern mental health. In turn, the screening results can be used by training programs to identify at-risk medical interns and connect them to appropriate clinical services [7,8,15,23,45,46].

Of additional note, male interns in our training set study sample were more likely to develop suicidal ideation. Although previous research suggests that female interns have higher rates of depression, [5] suicide fatalities appear to be more prevalent among males in both the general population [59] and among physicians [2,60,61]. Thus, our observed results underscore the greater risk of male suicide fatality. However, future research should continue to explore possible differences in suicidal ideation by intern demographics to most effectively identify groups at highest risk.

## Limitations

As with any observational study, we do not know if the factors we identified as being associated with increased risk of suicidal ideation are causal for suicidal ideation, or if they are the result of other unmeasured causal factors; we do know that they help predict suicidal ideation risk in these cohorts of interns. In addition, even though we were able to generally identify individuals with suicidal ideation, there were a subset of interns in our data with low predictive risk scores that also reported suicidal ideation within internship. Thus, future research should continue to evaluate predictors of suicidal ideation and identify possible reasons for underestimation of ideation risk.

Measurement error is one potential explanation for the underestimation of suicidal ideation risk in a subset of interns. We assessed suicidal ideation (and model predictor variables) through self-report inventories rather than diagnostic interviews. We chose this method, as opposed to an in-person assessment, based on previous data demonstrating that anonymity is

necessary to accurately ascertain mental health problems among medical students [62]. Nonetheless, it would be important to validate these findings using structured clinical interviews. Self-reports for predictor variables such as medical errors, work hours, and sleep hours could also lead to potential measurement bias, and thus future research should continue to explore different methods of assessment.

Furthermore, our study assessed suicidal ideation (using the ninth question of the PHQ-9) and not the much rarer outcomes of suicide attempts or suicide fatalities. The PHQ-9 question is broadly written to have strong sensitivity (but not necessarily specificity) for the measurement of passive suicidal ideation (i.e., thoughts of death) and self-injurious ideation. Therefore, we do not know if the risk factors we identify for suicidal ideation, based on the PHQ-9 response, are risk factors for physician suicide fatality. However, the Interpersonal Theory of Suicide states that suicidal ideation is a key determinant of suicide fatality risk [15], and a positive response to the PHQ-9 question has been shown to increase the cumulative risk for a suicide attempt or suicide fatality over the next year by 10- and 100-fold, respectively [30]. Recent recommendations to create and maintain a database for tracking medical student suicides [9] could help future research estimate the effects of risk factors on these rarer outcomes.

Lastly, given that our research was framed to potential participants as a study of depression, it is possible that interns with depression or suicidal ideation were more likely to participate. In this scenario, our study sample could have a greater prevalence of depression or suicidal ideation than the general population of interns, thus biasing our results. It is also possible that our study sample differs from the general population of interns across other demographics. However, previous comparisons of Intern Health Study participants and non-participants have only shown statistically non-significant differences in demographic variables (e.g., age, gender, specialty, institution) [63].

## Conclusions

The Interpersonal Theory of Suicide and empirical evidence from previous research suggest that physicians could have an elevated risk of suicide [14–20]. In this multi-site, prospective cohort study, we identified that a substantial proportion of training physicians developed suicidal ideation during their internship year. Factors assessed prior to and during internship had good predictive ability to identify the development of suicidal ideation and could be useful in clinical interventions to reduce the risk of suicide fatalities. In particular, self-reported work hours, medical errors, and sleep hours were identified as potentially mutable characteristics showing associations with suicidal ideation. Furthermore, interventions to address depressive symptoms could provide additional utility in the prevention of suicide. These predictors can empower interns and programs to prospectively understand suicidal ideation risk and take steps to mitigate risk before and during training.

## Supporting information

**S1 Fig. Risk of suicidal ideation during internship across all observations by prediction model.** *Notes*: Risk curves calculated for 2015 cohort test set by applying prediction models of suicidal ideation during internship constructed from 2012–2014 cohorts training set. For each risk curve, observations are ordered from highest risk of SI during internship to lowest risk of SI during internship. The rug plot underneath each risk curve indicates observations with suicidal ideation. SI = suicidal ideation; BASE = Model includes baseline predictors of suicidal ideation; BASE+PRIOR = Model includes base + prior quarter predictors of suicidal ideation; BASE+PRIOR+CUR = Model includes base+prior+current quarter predictors of suicidal ideation; No SI Covariates = Model includes base+prior+current quarter predictors of suicidal

ideation except for baseline and prior quarter suicidal ideation.
(PDF)

**S1 Table. Descriptive univariable analysis of suicidal ideation during internship and its association with intern demographics and baseline mental health, 2012–2014 cohorts training set.** *Notes*: Participants included in the above table were incoming first-year resident physicians (interns) that were assessed through the prospective cohort Intern Health Study. All interns in the table had complete baseline data and data for all tested explanatory variables over one set of consecutive internship quarter-intervals. Intern characteristics were self-reported. SI = suicidal ideation; SD = standard deviation. [a]No reported SI during internship. [b]Number of unique subjects in the given data set. [c]p-value for Pearson's chi-squared test of independence. [d]Assessed via the NEO-Five Factor Inventory. [e]p-value for the Satterthwaite two-sample *t*-test. [f]Assessed via the Patient Health Questionnaire-8. [g]Assessed via the General Anxiety Disorder-7.
(PDF)

## Acknowledgments

The authors wish to thank the interns who participated in the study.

## Author Contributions

**Conceptualization:** Zhou Zhao, Peter X. K. Song, Srijan Sen, Laura J. Scott.

**Data curation:** Tyler L. Malone, Zhou Zhao, Tzu-Ying Liu.

**Formal analysis:** Tyler L. Malone, Zhou Zhao, Tzu-Ying Liu, Peter X. K. Song, Srijan Sen, Laura J. Scott.

**Funding acquisition:** Srijan Sen.

**Investigation:** Zhou Zhao, Srijan Sen.

**Methodology:** Tyler L. Malone, Zhou Zhao, Tzu-Ying Liu, Peter X. K. Song, Srijan Sen, Laura J. Scott.

**Project administration:** Tyler L. Malone, Zhou Zhao, Tzu-Ying Liu, Peter X. K. Song, Srijan Sen, Laura J. Scott.

**Resources:** Srijan Sen.

**Software:** Tyler L. Malone, Zhou Zhao, Tzu-Ying Liu.

**Supervision:** Peter X. K. Song, Srijan Sen, Laura J. Scott.

**Validation:** Tyler L. Malone, Zhou Zhao, Tzu-Ying Liu, Peter X. K. Song, Srijan Sen, Laura J. Scott.

**Visualization:** Tyler L. Malone, Zhou Zhao, Tzu-Ying Liu, Peter X. K. Song, Srijan Sen, Laura J. Scott.

**Writing – original draft:** Tyler L. Malone, Zhou Zhao, Tzu-Ying Liu, Peter X. K. Song, Srijan Sen, Laura J. Scott.

**Writing – review & editing:** Tyler L. Malone, Zhou Zhao, Tzu-Ying Liu, Peter X. K. Song, Srijan Sen, Laura J. Scott.

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
