## [Decision Letter · Decision Letter 0]

31 Aug 2021

PONE-D-21-16857

Prediction of suicidal ideation risk in a prospective cohort study of medical interns

PLOS ONE

Dear Dr. Scott,

Thank you for submitting your manuscript to PLOS ONE. After careful consideration, we feel that it has merit but does not fully meet PLOS ONE’s publication criteria as it currently stands. Therefore, we invite you to submit a revised version of the manuscript that addresses the points raised during the review process.

The reviewers' comments are below.  I would echo the need for additional consideration of risk factors in the discussion. Additionally, the manuscript includes very little literature review in the introduction. The specific risk factors that were measured are not justified, and there is no consideration of theories of suicide risk, such as Joiner's interpersonal theory or other recent ideation-to-action frameworks. Inclusion of these, and interpretation of results in this context, would greatly strengthen the paper.

We look forward to receiving your revised manuscript.

Kind regards,

Neal Doran

Academic Editor

PLOS ONE

Journal Requirements:

I have read the journal’s policy and the authors of this manuscript have the following competing interests: SS reports having received research funding from the NIMH (R01 MH101459; website: https://www.nimh.nih.gov/index.shtml) and an American Foundation for Suicide Prevention Standard Research Grant (website: https://afsp.org/).  The funders had no role in study design, data collection and analysis, decision to publish, or preparation of the manuscript. 

Reviewers' comments:

Reviewer's Responses to Questions

**Comments to the Author**

1. Is the manuscript technically sound, and do the data support the conclusions?

Reviewer #1: Yes

Reviewer #2: Yes

2. Has the statistical analysis been performed appropriately and rigorously? 

Reviewer #1: Yes

Reviewer #2: Yes

3. Have the authors made all data underlying the findings in their manuscript fully available?

Reviewer #1: Yes

Reviewer #2: Yes

4. Is the manuscript presented in an intelligible fashion and written in standard English?

Reviewer #1: Yes

Reviewer #2: Yes

5. Review Comments to the Author

Reviewer #1: I think that a theory explaining the risk of suicide on medical students, interns, résidents... is lacking

(see the interpersonal theory on suicide and the work on physicians or medicla students)

cite and discuss (Cornette et al 2009; Fink-Miller 2015, Loas et al PLOS, 2018, 2019).

Reviewer #2: This is an important and well written paper that presents the overall results well in the abstract, but, oddly, does not include any substantive discussion about the risk factors identified in the training set (high neuroticism, personal history of depression etc) in the discussion, apart from the male demographic discussion. These merit at least a detailed paragraph of discussion - what is the potential biological significance/vulnerability of a past history of depression and high neuroticism and could these be screened for in addition to using tools like the phq9? Can such fairly easily identified risk factors be used by training programs? The authors also mention "second victim syndrome" in the discussion with reference to long hours of work and medical errors - but there is an increasing literature describing the impact of trauma on residents (especially repeat trauma) which could explain some of the longitudinal results presented, and a recent paper (Yellowlees et al) also described a link between Adverse Childhood Experiences (ACE'S) and burnout, which might be a more specific background reason and vulnerability than "neuroticism" as measured in this study.

6. PLOS authors have the option to publish the peer review history of their article (what does this mean?). If published, this will include your full peer review and any attached files.

Reviewer #1: No

Reviewer #2: No

---

## [Author Response · Author response to Decision Letter 0]

14 Oct 2021

Please see attached file, "Response to Reviewers.docx."

---

## [Decision Letter · Decision Letter 1]

28 Oct 2021

PONE-D-21-16857R1Prediction of suicidal ideation risk in a prospective cohort study of medical internsPLOS ONE

Dear Dr. Scott,

Thank you for submitting your manuscript to PLOS ONE. After careful consideration, we feel that it has merit but does not fully meet PLOS ONE’s publication criteria as it currently stands. Therefore, we invite you to submit a revised version of the manuscript that addresses the points raised during the review process.

This is an excellent study overall and the sample size and longitudinal nature of it are substantial strengths, as is the use of one cohort to develop a model that was then tested in a second cohort. There are a few minor concerns remaining. One is that while this is an observational design the manuscript includes some causal language. Please revise these to clarify what can be concluded. A few other necessary revisions are listed below.abstract, line 43: "...increased the risk of suicidal ideation..."abstract, line 48: "...leading to more medical errors..."page 14, line 335: "...that increased the..."in the header for Table 2, please clarify the timing of the SI outcome.Finally, statistical significance is binary rather than continuous. On page 11, line 249, please revise either to indicate that these were the significant predictors or the strongest predictors based on effect size (OR), rather than the "most significant" predictors.

We look forward to receiving your revised manuscript.

Kind regards,

Neal Doran

Academic Editor

PLOS ONE

Journal Requirements:

Reviewers' comments:

Reviewer's Responses to Questions

**Comments to the Author**

1. If the authors have adequately addressed your comments raised in a previous round of review and you feel that this manuscript is now acceptable for publication, you may indicate that here to bypass the “Comments to the Author” section, enter your conflict of interest statement in the “Confidential to Editor” section, and submit your "Accept" recommendation.

Reviewer #1: All comments have been addressed

Reviewer #2: All comments have been addressed

2. Is the manuscript technically sound, and do the data support the conclusions?

Reviewer #1: Yes

Reviewer #2: Yes

3. Has the statistical analysis been performed appropriately and rigorously? 

Reviewer #1: Yes

Reviewer #2: I Don't Know

4. Have the authors made all data underlying the findings in their manuscript fully available?

Reviewer #1: (No Response)

Reviewer #2: Yes

5. Is the manuscript presented in an intelligible fashion and written in standard English?

Reviewer #1: Yes

Reviewer #2: Yes

6. Review Comments to the Author

Reviewer #1: The remarks have been taken into account and thus the manuscript can be accepted without additionnal changes.

Reviewer #2: (No Response)

7. PLOS authors have the option to publish the peer review history of their article (what does this mean?). If published, this will include your full peer review and any attached files.

Reviewer #1: No

Reviewer #2: **Yes: **Peter Yellowlees

---

## [Author Response · Author response to Decision Letter 1]

8 Nov 2021

Please see the attached file, "Response to Reviewers.docx."

---

## [Editor Report · Decision Letter 2]

15 Nov 2021

Prediction of suicidal ideation risk in a prospective cohort study of medical interns

PONE-D-21-16857R2

Dear Dr. Scott,

We’re pleased to inform you that your manuscript has been judged scientifically suitable for publication and will be formally accepted for publication once it meets all outstanding technical requirements.

Kind regards,

Neal Doran

Academic Editor

PLOS ONE
---

## [Editor Report · Acceptance letter]

22 Nov 2021

PONE-D-21-16857R2 

Prediction of suicidal ideation risk in a prospective cohort study of medical interns 

Dear Dr. Scott:

I'm pleased to inform you that your manuscript has been deemed suitable for publication in PLOS ONE. Congratulations! Your manuscript is now with our production department. 

Kind regards, 

on behalf of

Dr. Neal Doran 

Academic Editor

PLOS ONE